# Mobile Phone Data: A Survey of Techniques, Features, and Applications

**DOI:** 10.3390/s23020908

**Published:** 2023-01-12

**Authors:** Mohammed Okmi, Lip Yee Por, Tan Fong Ang, Chin Soon Ku

**Affiliations:** 1Faculty of Computer Science and Information Technology, University of Malaya, Kuala Lumpur 50603, Malaysia; 2Department of Information Technology and Security, Jazan University, Jazan 45142, Saudi Arabia; 3Department of Computer Science, Universiti Tunku Abdul Rahman, Kampar 31900, Malaysia

**Keywords:** mobile phone data, call detail records (CDRs), mobility patterns, communication behaviors, urban crime patterns, urban sensors, smartphones

## Abstract

Due to the rapid growth in the use of smartphones, the digital traces (e.g., mobile phone data, call detail records) left by the use of these devices have been widely employed to assess and predict human communication behaviors and mobility patterns in various disciplines and domains, such as urban sensing, epidemiology, public transportation, data protection, and criminology. These digital traces provide significant spatiotemporal (geospatial and time-related) data, revealing people’s mobility patterns as well as communication (incoming and outgoing calls) data, revealing people’s social networks and interactions. Thus, service providers collect smartphone data by recording the details of every user activity or interaction (e.g., making a phone call, sending a text message, or accessing the internet) done using a smartphone and storing these details on their databases. This paper surveys different methods and approaches for assessing and predicting human communication behaviors and mobility patterns from mobile phone data and differentiates them in terms of their strengths and weaknesses. It also gives information about spatial, temporal, and call characteristics that have been extracted from mobile phone data and used to model how people communicate and move. We survey mobile phone data research published between 2013 and 2021 from eight main databases, namely, the ACM Digital Library, IEEE Xplore, MDPI, SAGE, Science Direct, Scopus, SpringerLink, and Web of Science. Based on our inclusion and exclusion criteria, 148 studies were selected.

## 1. Introduction

In the past few decades, mobile phone data has grown into a stand-alone topic [1]. Due to the wide use of smartphones [2], the digital traces left by smartphone use have come to provide valuable real-time information about human activities. These digital traces facilitate the study of human behaviors. Various techniques and analytical perspectives have been employed to capture many aspects of human behaviors from mobile phone data, which has resulted in various applications. These analytical perspectives, such as human mobility, communication patterns, social networks, and mobile phone usage activities, have been built on various spatiotemporal and call characteristics extracted from mobile phone data.

For example, in the context of criminology, refs. [3,4] constructed a criminal network based on the communication traces that had been left by criminals over a period of time. The traces included details about the locations where the criminals had received calls, the timestamps of their communications, the frequency of their calls, and the periods of their activity. When these traces were collected over a period of time and put together, they provided insights into the communication patterns of the individuals concerned, which were then used to infer their criminal behavior and relationships. Griffiths et al. [5] analyzed the mobility traces of criminals based on the digital traces they had left at their homes and at other meaningful locations (e.g., the crime scene) to determine whether the criminals’ movements were regular or random. The researchers subsequently concluded that there was a high degree of spatial regularity in the criminals’ movements. In the field of urban sensing, the same process has been used for a number of other things, such as figuring out people’s social ties and interactions [6] and figuring out how the population changes every day [7].

Spatiotemporal mobility patterns and mobile phone activity patterns have also been widely extracted from mobile phone data for other practical applications, such as capturing individuals’ activities associated with urban zones, transportation, and the COVID-19 pandemic. The authors of [8,9,10,11,12], among others, explored or captured human activity patterns from mobile phone data to infer land use types based on spatiotemporal call volume patterns. This feature represents the total call volume (the number of incoming calls received, and outgoing calls made from all smartphones) managed by a given base transceiver station (BTS) within a given period of time. On the other hand, Mao et al. [13], Lenormand et al. [14], and Ríos and Muñoz [15] inferred land use based on temporal changes in human activities. For example, Ríos and Muñoz [15] detected land use patterns based on changes over time (temporal changes) in human activities. They did this by extracting the spatiotemporal features from mobile phone data, such as the number of calls managed by every BTS every hour per week. Empirically, to identify human behavior patterns based on temporal changes, the activity pattern (total number of calls) of each BTS in a week (from Sunday to Saturday and every hour) is depicted by the number of calls managed by the BTS every hour over a seven-day week. This results in a list of consecutive time frames (time windows) [15] or creates a 7-day/24-h call pattern time series.

Another example of practical application of extracting mobile phone activity patterns from mobile phone data is mapping population distributions based on mobility patterns that shape human presence and mobility in space and time [16] and allow for understanding the spatiotemporal dynamics of an urban population [17]. Here, multiple spatiotemporal features have been extracted to depict human mobility patterns. For example, the cell tower ID shows the geographical location of the smartphone from which a call was made or received (smartphones connect regularly to the nearest cell towers), along with a timestamp record of when the interactive event happened. It also shows whether users are more active during the daytime or nighttime [17], based on when the calls were made.

We noticed that the human activity characteristics extracted from mobile phone data varied from one study to another. The concept of circadian rhythm in human activities has been empirically depicted in different ways. Thus, as mentioned earlier, this survey was conducted to provide a detailed description of the multiple features extracted from mobile phone data to depict human activities. Furthermore, the extant literature provides various processing techniques and analytical perspectives to capture and depict different aspects of human behavior, such as human mobility patterns, social networks, and communication behaviors. As mentioned earlier, these analytical perspectives have been built based on multiple spatiotemporal and call characteristics and features extracted from mobile phone data. Additionally, different types of mobile phone data show that different spatiotemporal and call characteristics can be extracted to map people’s behaviors. The goal of this survey was to figure out what these characteristics are and what they do.

In short, this survey looks at the aspects of human behavior (mobility, communication, and social networks) that can be learned about by analyzing mobile phone data. This includes the spatiotemporal and call features of the data, which show patterns in how people move and talk to each other, as well as the methods that have been used to analyze mobile phone data to learn about these things.

The primary objective of this review is to investigate the current state of mobile phone data applications in various domains, with a specific focus on topics that the current literature has not covered, especially in the fields of criminology and urban sensing. Although Blondel et al. [1], Ghahramani et al. [2], and Calabrese et al. [18] made impressive contributions by examining mobile phone data applications in several studies, there is still a lack of research relating to mobile phone data in crime and land use. For example, Blondel et al. [1] reviewed social network applications that can be derived from mobile phone data in various disciplines and domains, such as urban sensing, epidemics, public transportation, and data protection. In addition, Ghahramani et al. [2] and Calabrese et al. [18] presented a survey of mobile phone data applications in the urban sensing domain. However, to the best of our knowledge, this is the first survey on the use of mobile phone data in crime and land use applications. No one had fully reviewed or discussed criminal applications for mobile phone data. This paper’s key contributions are its discussions of various techniques and analytical perspectives that have been derived from mobile phone data to capture many aspects of human behavior, such as communication, social networking, and mobility patterns. The review also places a heavy focus on human attributes and characteristics that can be derived from mobile phone data by explaining the spatiotemporal and call features that have been extracted from mobile phone data to model human mobility and communication patterns and differentiate their functionalities. The review also sheds light on anomaly detection, churn prediction, and crime prediction applications, other topics that the literature lacks.

The remainder of the paper is organized as follows: in Section 2, we present the research methodology. In Section 3, we describe different types of mobile phone data. Mobile phone data applications in urban crime and urban sensing are discussed in Section 4. Section 5 discusses mobile phone social network applications. Section 6 discusses publicly available datasets, managerial implications, and methods. Section 7 provides potential research opportunities. Section 8 discusses privacy concerns, and Section 9 concludes the review.

## 2. Survey Methodology

This study reviews state-of-the-art methods and techniques regarding the use of mobile phone data in crime and urban research to generate the focus of the survey. In addition, the survey aimed to shed light on related studies or contributions made in mobile phone data research that the literature lacks, such as anomaly detection, churn prediction, and privacy concerns across various disciplines. Therefore, the survey was conducted using eight databases to search for relevant journal articles and conference papers. These sources include the following databases and digital libraries: ACM Digital Library, IEEE Xplore, MDPI, SAGE, Science Direct, Scopus, SpringerLink, and Web of Science. We ran keyword searches against titles, abstracts, or keywords in these data sources, allowing us to select the most relevant research journals and conference proceedings. The main keywords we used for our searches were “mobile phone data,” “call detailed records,” “call detail records,” “call data records,” “mobile phone datasets,” “mobile phone networks,” “mobile phone network data,” “mobile network data,” “mobile network activity,” “mobile communication data,” and “mobile phone call detail records.” After that, we filtered the results for articles published between 2013 and 2021 according to our inclusion and exclusion criteria. This allowed us to choose 148 articles for review. The following are the inclusion criteria (IC) and exclusion criteria (EC) that were used to decide which articles to choose and which ones to leave out.

IC1: A study has to be in a journal or proceedingsIC2: Studies are peer-reviewed articlesIC3: A study must be written in the English languageIC4: A study must be published from 2013 to 2021EC1: Articles that are not written in EnglishEC2: A study that is not published between 2013 and 2021

## 3. Mobile Phone Data Types

According to the literature [1,18,19], there are two types of mobile phone data: a type that records the details of an interaction between a mobile device and the network, known as event-driven mobile phone data, and another type, based on the cell tower location updates of mobile phones, known as network-driven mobile phone data.

In the first type, the data records the details of communication events that occur when mobile phones receive or initiate phone calls, text messages, or access the internet. The details of each communication event are held in a record that includes the IDs of the caller and the callee, the caller’s connected cell tower ID, the callee’s connected cell tower ID, the duration of the call, and a timestamp. Thus, this type is called “event-based mobile phone data,” and it can be formed at individual and group (aggregated) levels. At the individual level, the data convey details about the communication and mobility behaviors of each party involved in the call; and at the group level (where the data are aggregated based on grouping a certain number of users in a certain area based on their connection to the cellular tower at various spatial and temporal scales), the aggregated data can provide relevant details including the user ID, the timestamp, and the cell tower ID for each of the individuals who are involved in the call.

The aggregated mobile phone data (usually called aggregated CDRs (call detail records), or mobile phone data at aggregated level) differs from the individual data (usually called mobile phone data at the individual level, or CDRs data) in the sense that they are anonymized and aggregated at different spatial and temporal scales. Thus, they are easier to collect and acquire, and their management is simpler compared to the individual data because they are aggregated based on the mobile phone devices that are connected to the BTS at different spatial and temporal scales. As a result, this allows for: the observation and monitoring of the spatial and temporal fluctuations of the population’s activities at various scales (e.g., hourly, daily, seasonal) based on their movement patterns; the classification and clustering of users by parameters derived from location information, such as the most visited places or the most frequent locations that the users are present in; or to distinguish the residents of a place from visitors based on calling volume (the number of phone calls) that can depict the amount of activity in a given location. All of these measures can be acquired from the aggregated CDR data that offers spatiotemporal mobility patterns and communication behaviors. In contrast, the non-aggregated mobile phone data at the individual level (CDRs) requires permission from authorities for their collections and is difficult to manage due to the raw nature of the data, which requires cleaning and preprocessing steps before the data can be fully utilized. In addition, the raw data are hard to understand, and you need data mining tools to get useful information out of it.

Alternatively, the second type of mobile phone data aggregated at the cell tower level (which is also called “location-based mobile phone data”, “passive mobile phone records”, or “network-driven mobile phone data”) primarily involves storing location information updates of mobile devices based on cell tower locations. In this type, the data records passive location updates of a mobile device from the base transceiver station (BTS) (also known as a cell, cell tower, or node B in 3G networks) at regular intervals and whenever the mobile device is switched on or off, receives a signal from the mobile network, or changes the type of connection. This type of data stores a record that details every event, including the user’s ID, the timestamp, and the location (the cell ID), where each record is geolocated based on the nearest BTS the mobile device is connected to. Thus, in every geographical area, there is a defined number of BTS that cover the given area to ensure the quality of the communication services and optimize the signaling. Each of the mobile network events is recorded to a BTS through a base station controller (BSC) that controls and manages a set of BTS by governing the network traffic and performing network handover. Accordingly, whenever the signal between a mobile device and a connected BTS becomes weak, the BSC will hand over the connection to another BTS to guarantee an optimal transmission signal for that connection. This type differs from the first one in the sense that it neither depends on active events initiated by the user nor does it reveal communication (calling) information about the receiving device and its location (i.e., the terminating user ID and the cell ID); thus, mobile phone data at cell tower level mostly captures spatial–temporal human mobility patterns due to the fact that only spatiotemporal information is recorded. Additionally, this type of data can record events either actively or passively based on how the mobile device interacts with the cell towers (passively, when the phone is turned on or off, receives a signal from the mobile network, changes the type of connection, etc.; actively, when a call is made, a text message is sent, etc.).

## 4. Human Mobility Patterns

Spatiotemporal information provided by mobile phone data can provide clearer insights into human movements and interactions in various applications. For instance, mobile phone data have been used to detect certain types of behaviors in cities and urban zones based on human mobility patterns, to investigate the relationship between human dynamics and crimes, to estimate population density, and to detect home and work locations. Therefore, analyzing human dynamics and movements is crucial to understanding their actions and activities.

### 4.1. Urban Environment

With the rapid growth and usage of mobile devices in urban environments, mobile phone data have been widely used as urban monitoring sensors, resulting in many diverse applications such as land use classification, estimating population density, home-to-work identification, urban hotspot detection, and others. So, looking into urban sensing applications is a good way to find out where and how much people do things in cities [2].

### 4.2. Classification of Urban Land Use

In mobile phone data studies, land use inference is based on human dynamics and activities as extracted from their mobility and call patterns. Spatiotemporal information extracted from human mobility patterns can, in this manner, be used to depict human activity patterns that shape land use types, with human activity patterns assumed to be linked to the time and location of recorded mobile phone usage based on the locations in which such activities take place [20]. Because of this, people’s mobility and movement can stand in for many of the things they do, which can then be used to figure out how they use the land.

Empirically, human activity patterns based on mobile phone data are derived from specific characteristics provided by such data, specifically the temporal and spatial characteristics of mobile phone users as represented by concentrations of locations and times of mobile activities; for example, more activities occurring during the day (8 a.m. to 6 p.m.) on weekdays may imply that the main land use is for commercial or business areas, due to the fact that these types of areas have higher concentrations of mobile activities. Such human activity patterns as extracted from mobile phone data have been used to represent human activity patterns to infer land use over several studies, with most feature types being used, including spatiotemporal calling volume pattern features. This allows the capture of the hourly patterns of human activities across a given period of time. Human activity and mobility patterns are thus assumed to be reflected in call volume patterns, and such patterns can be captured when users make phone calls; their locations are recorded [21] and can thus be used to identify land use. Several studies have thus examined different features extracted from mobile phone data and used varying approaches to infer land use based on human activities.

Being able to capture human daily activities based on mobile phone data allows for basic classification of land use areas. Thus, several studies in this application have intended to apply classification and clustering methods to identify land use types. A semi-supervised clustering fuzzy c-means (FCM) method was proposed to classify urban land use in Singapore into five land use types: residential, business, commercial, open space, and other [8]. The authors constructed a synthesized vector from a linear combination of two real vectors. These two vectors represent human activity patterns extracted from mobile phone data, showing the total calling volume and hourly call volume managed by each BTS. This was intended to overcome the gap left in previous studies where the time series of the call volume for two-day patterns were considered to infer land use types. The semi-supervised FCM was then used to classify the urban land use based on the synthetic vector. The same features were extracted from mobile phone data by [9] to represent human activities; however, they used a different approach whereby an unsupervised algorithm known as non-negative matrix factorization (NMF), which is well known for its use in dimensional reduction and feature extraction, was applied. The authors thus decomposed the call pattern matrix A, which contained only non-negative coefficients representing human activity patterns, into the product of two matrices, W and H (also composed of non-negative coefficients). After decomposing the call pattern matrix in this way, two basis vectors were identified, which illustrate the commercial/business/industrial (C/B/I) (work pattern) and residential (out-of-work pattern) characteristics of various areas. Mao et al. [13] used a similar approach to identify land use types with an extra feature extracted to depict human activities more precisely: temporal mobile phone call patterns were thus extracted to reveal different land uses along with total call volumes as aggregated at each cell tower. These two feature types helped to distinguish between those human activities that varied from weekdays to weekends. For example, the amount of time people spend at home during weekdays, based on features generated to represent the call volume by the hour of the day for each day of the week, showed that human activity in residential areas is up to three times higher at weekends due to the fact that people go to work on weekdays and stay at home more during weekends. Thus, during weekdays, activity reaches a peak in the middle of the day, between 12 p.m. and 6 p.m. in business areas. Yuan et al. [12] presented an unsupervised k-means clustering model that aimed to identify urban functional areas (UFAs) based on citizens’ daily activities and communication activity intensity. To model this communication activity intensity (CAI), various spatial and temporal characteristics were extracted from mobile phone data, including the ratio of the number of calls made or received in a specific geographical area subdivision (GASs) during a specific time interval. Lastly, Furno et al. [22] used an unsupervised signature clustering algorithm to try to improve land use classification by combining mobile phone data with additional mobility information taken from GPS data, specifically the GPS traces of cars that were floating.

Using a supervised learning model to classify land use has been attempted by [11] and [23], with a sample of land use initially labeled to train the classifier. Zinman and Lerner [23] applied a random forest (RF) algorithm to classify urban areas in Tel Aviv by extracting two types of features previously used across the literature [8,9,10,12,13] along with additional feature types representing communication habits to capture human communication behaviors, such as call duration, contact type (phone calls, accessing the internet), the weekly pattern features based on differences in communication activity between weekdays and weekends, and contact features such as the average number of days on which people engaged in or performed cellular contact in a given cell over one hour. Similarly, [11] applied a support vector machine (SVM) classifier algorithm, one of the most popular supervised learning algorithms, to classify urban land use types in Beijing. These were split into six classes: (a) residential, (b) business, (c) scenic, (d) open, (e) other, and (f) entertainment. As SVM is a supervised ML approach, it requires training datasets with land use labels to train the model. Thus, based on the fact that mobile phone data are not inherently labeled data, the authors selected land use labeled samples from Google Earth™ at a resolution of 2776 pixels as the training samples to train the appropriate classifiers. During the process of putting together human activity patterns from mobile phone data, the authors used the same spatiotemporal characteristics to show human activity patterns as those most often used in previous studies [8,9,10,12,13].

A different approach based on community detection techniques (community graph clustering), which is different from traditional clustering techniques such as K-means, FCM, and hierarchical cluster analysis (HCA) to detect land use types, was adopted by [14]. Lenormand et al. [14] wanted to make activity profiles based on the weekly activity that show the number of mobile phone users per hour during the days of the week to find four land use types (residential, business, logistics/industrial, and nightlife) in Spain.

As a way to improve land use classification outside of classical clustering techniques, Ríos and Muñoz [15] proposed a new clustering technique that aims to detect land use in Santiago City. This was based on adopting a latent variable clustering technique.

Other studies have attempted not only to classify land use but also to investigate additional information about the relationships between human activities and urban land use and how different urban land use types can influence or impact people’s lives. Jacobs-Crisioni et al. [20] aimed to investigate the impact of mixed and dense land use on urban activity dynamics in Amsterdam, Netherlands, to determine whether land use density and mix can prolong high levels of activity in urban areas; they thus extracted spatiotemporal characteristics from mobile phone data to depict human activity levels and then applied spatial regression models. The results showed that urban areas with high attraction levels corresponded with increased urban activity intensity, supporting the hypothesis that mixed land use diversifies urban activity dynamics and increases profiling activity intensity. Put simply, mixing shops, businesses, and meeting places has an additive effect on activity levels. Yang et al. [24] aimed to explore the relationship between human convergence and divergence patterns extracted from mobile phone data and land use characteristics (commercial, industrial, residential, public, and transport land). To achieve such a goal, the authors first extracted spatial and temporal characteristics provided by mobile phone data to depict human convergence–divergence patterns and then applied multinomial logistic regression (MLR) to reveal the effects of land use characteristics on human convergence–divergence dynamics. Liu et al. [25] used commuting flow patterns (origin–destination (OD) commuting flows) to depict human activities with the aim of exploring urban land use types as well as investigating whether urban land use influences commuting flows. By using a Louvain modularity-based algorithm, Novovic et al. [26] attempted to apply community detection techniques based on users’ dynamics and activity variations over space and time to infer the correlations between human dynamics and land use. The idea of modularity has been used a lot in research on mobile phone data as a good way to measure how good a partition is.

The variation in researchers’ findings on land use can be attributed to different perspectives on how to analyze mobile phone data in order to depict human activities. For example, individuals’ behaviors have been interpreted in diverse ways, including, among others, user dynamics, human activities and mobility patterns, commute flow patterns, temporal activity patterns, and human convergence/divergence dynamics. These multiple analytical perspectives have been developed to deal with the diverse spatiotemporal and call features extracted from mobile phone data, such as spatiotemporal call volume, users’ daily trajectories, communication habits, weekly patterns, and contact features.

Table 1 shows the different analytical perspectives and features extracted, as well as the methods used to infer land use. Previous studies have shown that detecting land use is best handled as a clustering problem to which algorithms can be applied, while other statistical methods have been used to uncover the relationships between human activities and land use.

### 4.3. Urban Crime Research

Investigating densely populated urban environments where criminal activities are much more likely to occur has long been a popular topic in mobile phone data research. The structure of built urban environments can be seen in many attractions and centers of activity such as bars, clubs, hotels, shops, and schools that draw many people and create opportunities to commit crimes as these places are known for being crowded. Urbanization pulls multiple individuals to cities and thus results in more crime [27]. Empirical research has shown that urban environments where people concentrate and participate in daily activities, for example, schools, stations, shopping centers, sports venues, and entertainment areas, can be defined as crime generators. Crime attractors, in turn, are places that are not crowded but attract people who want to commit crimes [28], such as bars, nightclubs, automatic teller machines, and banks [27].

Mobile phone data have been used to prevent, fight, and deter urban crime by predicting crime hotspots, detecting and identifying suspects and criminals, and investigating the relationships between human mobility patterns and criminal activity patterns. The digital tracks left by people at locations where a crime has taken place can reveal a representative sample of the population present at the crime scene at a given time, thereby providing insights into correlations between individuals’ mobility and criminal activities. Mobile phones connect to a given cell tower located in the area where a crime has taken place, leaving digital traces or evidence that can be used to conduct empirical research on crime patterns and the dynamics of criminal behavior from a people- and place-centric perspective.

The literature reports various crime-related applications that have been derived from mobile phone data aggregated at the cell tower level. For example, some studies [29,30] have identified crime hotspots in London by extracting human mobility patterns from mobile phone records. To depict these mobility behaviors, the cited authors focused on spatiotemporal features, such as the total numbers of calls or mobile phone devices for each cell tower every hour. Other researchers [27,28,31,32,33] have similarly relied on defining specific populations’ spatiotemporal patterns to investigate the relationships between human dynamics and spatiotemporal crime patterns. These studies have tracked features of ambient populations that indicate specific areas are at risk of criminal activities or that individuals are in danger of becoming victims of theft or assault, thereby confirming that ambient populations’ configurations have a significant impact on crime patterns and rates.

Scholars have discovered additional applications by analyzing mobile phone data at the individual level, or CDRs. For example, previous studies [34,35,36,37,38] have detected or constructed criminal social networks based on criminals’ communication behaviors. To depict this type of communication, the cited authors have extracted call features such as the duration of calls between two criminals, the frequency with which they make calls, the maximum, average, and minimum duration of these calls, and the maximum number of outgoing phone calls or messages. These features then allow these researchers [34,35,36,37,38] to represent criminal organization networks that show whether nodes (i.e., criminals) are highly influential or less influential members. The results can, in turn, help criminal investigators understand criminal networks’ hierarchical structure, as criminal investigators have trouble determining who belongs to criminal organizations, who heads them, and what relationships exist within them due to the nature of the raw data available.

Still, other studies [4,39,40] have built forensic detection models to separate suspects from non-suspects. The cited authors have analyzed criminal communication behaviors by extracting suspects’ call features such as timestamp, frequency, and average, maximum, average, and minimum duration. Table 2 presents crime-related applications based on mobile phone data, the various features extracted to depict human behaviors, and the methods used.

As noted in Table 2, most researchers have used statistical methods such as correlation and regression analysis in crime prediction, and SNA tools have been widely used to detect criminal networks. In suspect identification, big data technologies are the most frequently used. Moreover, the variation in research on crimes can also be viewed from various analytical perspectives, and features extracted from mobile phone data can be used to depict human activities, as illustrated in Table 2 and Figure 1.

In addition, some studies applied spatial mapping techniques to intersect or project mobile phone locations from mobile network cells [53] into spatial units, such as census units [42] and Lower Layer Super Output Area (LSOAs) units [44,45].

However, other studies [48,49,50,51,52] have not considered the spatial mapping of mobile devices to a given spatial unit. In this case, mobile phone locations were originally assigned to the coverage areas of a base station [16]. Consequently, undoing the spatial mapping of mobile phone presence will affect the accuracy of the sampling population at or near crime scenes and can lead to inaccurate detection results.

### 4.4. Public Health

Mobile phone data have been used in public health research to fight against infectious diseases, understand human mobility after natural disasters, measure and estimate human mobility in relation to the epidemiology of infectious diseases, and quantify exposure to air pollution [54]. These above-mentioned multiple health applications have been derived from human mobility patterns by extracting spatiotemporal characteristics of individuals from mobile phone data. Indeed, many aspects of human daily activities and lifestyles are linked to human mobility patterns, and measuring and monitoring human mobility patterns can aid in combating the spread of infectious diseases and avoiding potential threats to public safety and human health. For example, [55] depicted individual mobility patterns by extracting their spatiotemporal features to examine and quantify their exposure to air pollution.

Recently, human mobility patterns and interactions have allowed a better understanding of COVID-19 trends and geographic distribution [56]. This is one of the important roles that mobile phone data has played recently in controlling and preventing COVID-19 and tracking population movements. During the coronavirus disease 2019 (COVID-19) pandemic, many COVID-19 applications were built using data from mobile phones. These applications were used, for example, to help public health response to COVID-19 [57], to track changes in people’s mobility patterns [58], to control how the COVID-19 pandemic spreads [59], and to find out if there is a link between changes in mobility patterns and the number of daily COVID-19 cases [60].

So, being able to measure and track people’s movements can help reduce health risks [61] and people’s exposure to air pollution, which can cause health problems such as breathing and heart problems [62]. Table 3 presents public health applications based on mobile phone data, analysis perspectives, and study findings.

As noted in Table 3, several applications have been examined in health research. It can be observed that travel and mobility patterns have been widely used to depict human activities, allowing investigation into the effect of travel and the transmission of infectious disease and estimating the number of trips to areas with a higher risk of COVID-19.

### 4.5. Transportation Research

Due to the fast growth of telecommunication networks, a large quantity of data about how people move and behave in space and time is being generated, which could be used to evaluate and analyze the travel patterns and social interactions of the total population [69]. Understanding the movement of people and where they live is essential because it can help urban planners and organizers manage traffic flow and plan public transportation services [2]. In this matter, mobile phone data have played an essential role in depicting people’s mobility and social interactions in urban areas due to the wide usage of smartphones. The digital traces (e.g., mobile phone data) left by this large number of devices provide valuable information that facilitates the understanding of passenger travel behaviors, human movement behaviors in urban areas, and changes in human driving behaviors during road traffic. Because of this, knowing how people travel and act could lead to ways to improve the quality of life in a city on a large scale.

However, one of the challenges in transportation research is to understand the travel behavior of passengers with regard to identifying the hub passengers and detecting their transport modes (e.g., train and subway). Analyzing the transportation hub, which is the transition point for passengers to switch between various types of transportation (train and subway), is necessary in order to understand passenger travel demand [70]. This ultimately helps to support and evaluate urban transportation planning and management [71].

The second challenge on which transportation literature widely focuses is enhancing the detection of the transport mode of passengers. This difficulty is due to the nature of mobile phone data, which is noisy, sparse, and irregular. Several techniques have been proposed to improve the performance of transport mode detection. For example, Graells-Garrido et al. [72] aim to identify the distribution of usage of transportation modes in Santiago, such as the metro, bus, and car, by building a pipeline method for inferring trips based on user trajectories. Chin et al. [73] aim to improve the detection of transport modes by applying multiple supervised and unsupervised detection algorithms such as rule-based heuristics, RF, fuzzy logic, and partitioning around medoids (PAM). The author in [74] aim to identify travelers’ transport modes, such as buses, cars, and railways, based on detecting travel speeds for each transport mode. For example, if a trip’s travel speed exceeds 15 km/h and there are no subway or bus stops within 500 meters of its source or destination, the trip is designated as a car trip [69]. Finally, Bachir et al. [19] aim to estimate total origin–destination flows by modes of transportation, such as road and rail, by combining five different types of mobility datasets that allow for fine-grained daily population dynamics.

There are also other problems, such as the fact that existing mode detection techniques are mostly made to recognize easy-to-detect modes (subway, train) or more general mode groups (e.g., rail versus road, moving versus stationary) [69,73].

The literature shows that mobile phone data have been used for many transportation research purposes, such as estimating passenger origin–destination flows, predicting traffic congestion, and other applications. Table 4 presents transportation applications.

## 5. Human Communication Behaviors

Mobile phone data at the individual and aggregated levels, usually called CDRs and aggregated CDRs, respectively, can be used to investigate and study human communication behaviors and social communication patterns due to the fact that mobile phone data at these levels contain communication information. Mobile phone data aggregated at the cell tower level does not contain details of the other side of communication (such as the callee ID or the callee’s connected cell tower ID) and thus cannot be used in applications regarding human communication behaviors. So, CDRs have been used to study a wide range of human social-related topics in many different ways. For example, it has been used to infer social ties and relationships, find social networking communities, find criminal relationships, identify suspects and criminals based on their calling patterns, and do other things that have to do with how people talk to each other.

### 5.1. The Construction of Social Networks

Mobile phone network data can be used to reconstruct social network interactions [1], which depict social structures as mobile phone customer interconnections. Most of the time, this is achieved through the use of mobile phone data as a graph consisting of a set of nodes and a set of edges.

Generally, any social network can be represented by a graph, *𝐺*, on a set of nodes, *𝑉*, and a set of edges, *𝐸*. To construct a social network of criminals based on communication behaviors extracted from CDRs, calling characteristics must be extracted, such as the number of calls or SMS messages, duration of calls, and timestamps. Empirically, mobile phone data can create a social network based on individuals (subscribers) making or receiving calls or messages; these individuals are classified as actors (nodes) within the network, and the links between the actors represent the various types of communications (e.g., calls or messages). In a real-life scenario detecting a criminal social network, Taha and Yoo [35,36] created a criminal social network derived from mobile phone data that showed a criminal network consisting of 62 nodes representing all individuals involved in the incident; this contained 153 edges, where these edges represented various reported interactions (such as calls and SMS messages) between the actors (criminals) involved in the incident. However, in studies of mobile phone data, different ways to build or make social networks have been found. These ways vary from one study to the next based on the purpose of the study and the features taken from mobile phone data.

In addition, different network metrics (see Section 4.2 for more details) may be applied during any network analysis, such as the use of a centrality measure or metric to find the most influential subscribers in the network or the use of a reciprocity measure [82] to analyze linking behaviors in a network by determining whether they demonstrate transitive behaviors between nodes [83]. Yet there are also other network metrics that are used for different things, such as figuring out how a network is put together or figuring out how strong the modules in a network are, such as network modularity and efficiency. Table 5 shows some studies that analyze mobile phone data based on social network analysis (SNA) tools.

### 5.2. Network Metrics

Degree centrality: The number of direct links that a node possesses can be defined as its degree of centrality. Any nodes with a high degree of centrality may be regarded as hubs (crucial channels of communication). Equation (1) can be used to calculate degree centrality, in which *n* represents the number of vertices in the network, whilst $x_{vu}$ will be equal to 1 if vertices *v* and *u* are linked. If not, this figure will be 0 [3].
(1)$$C_{I}\left(u\right)=\overset{n}{\underset{v\neq u}{\sum }}x_{vu}$$

Betweenness centrality: This metric assesses the distance between a specific node and other nodes in the same network. These intermediate elements can exert strategic control and influence on other elements [34]. This centrality measure is largely based on the assumption that any node lying between the shortest path connecting two nodes is central. People who have a high degree of centrality are thought to act as “gamekeepers” in the network.

Shortest paths: The number of shortest paths between vertices passing through the vertex can be used to calculate its betweenness centrality [3]. Equation (2) can be used to work out the betweenness centrality, in which $\sigma_{st}$ represents the number of shortest paths between vertex *s* and vertex *t*, whilst $\sigma_{st}\left(u\right)$*)* represents the number of shortest paths between vertex *s* and vertex *t* passing through *u*.
(2)$$C_{B}\left(u\right)=\underset{s,t\in G,s\neq t}{\sum }\frac{\sigma_{st}\left(u\right)}{\sigma_{st}}$$

Closeness Centrality: Closeness refers to the length of the shortest path compared to all other vertices. In other words, it assesses the proximity of a vertex to other vertices. Equation (3) can be used to measure closeness centrality, in which (*ui-*, *u*j) represents the distance between vertex *ui* and vertex *u*j.
(3)$$c_{c}\left(u_{i}\right)=\frac{1}{\sum_{j\neq i}^{n}d\left(u_{i},u_{j}\right)}$$

Identifying the influential members in criminal networks by assigning weight to a vertex: Here is an initial formula proposed by [36] that can be used to assign each vertex $v_{k}$ with a weight that accurately represents its importance in the network compared to other vertices. To calculate the weight of k, the following factors must be considered: (a) The frequency of incoming and outgoing communications to $v_{k}$ (i.e., the number of times vertex k appears in reports of criminal activity compared to other vertices). (b) The frequency of incoming and outgoing edges to $v_{k}$ (i.e., the number of vertices with edges outgoing to k, as well as the number of vertices with edges incoming from $v_{k}$). The following equation was employed by the researchers to calculate the relative weights.
(4)$$w\left(v_{k}\right)=\frac{\sum_{i=1}^{\left|v_{k}^{E}\left(in\right)\right|}\left|\left(v_{i},v_{k}\right)\right|+\sum_{j=1}^{\left|v_{k}^{E}\left(out\right)\right|}\left|\left(v_{k},v_{j}\right)\right|}{\sum_{i=1}^{\left|v_{k}^{E}\left(in\right)\right|}0.8\left|v_{i},v_{k}\right|+\sum_{j=1}^{\left|v_{k}^{E}\left(out\right)\right|}0.6\left|v_{k},v_{j}\right|}$$

## 6. Discussion

The study results show that various spatiotemporal and call features and characteristics have been extracted from mobile phone data to depict human activity patterns. These characteristics vary between studies based on their purposes and the aspects of human behaviors that they aim to investigate (i.e., mobility patterns, social interaction, communication, etc.). For example, to identify suspects, [48,50] extracted call features, such as phone calls that suspects made or received at crime scenes, using them as evidence of their involvement in the crimes. Similarly, [39] identified suspects based on call features but with additional features such as “average call duration with suspects,” “maximum call duration,” “average call duration,” “minimum duration of a call out,” “maximum duration of an incoming call,” and “standard deviation duration of an incoming call.” In other applications, [8,9] explored human activity patterns to infer urban land use based on spatiotemporal calling volume patterns. Similarly, [23] inferred urban land use based on spatiotemporal calling volume patterns and additional features, such as communication habits, weekly patterns, and contact features. Here, we notice that multiple features have been extracted to depict human behaviors, relying on either spatiotemporal or call features. However, there is currently no standardized way in which specific features should be extracted, as the literature demonstrates multiple approaches. There is also no scientific evidence on which spatiotemporal and call features are better than others when it comes to accurately and effectively reflecting on and depicting human behaviors.

In this section, we also discuss the availability of mobile phone datasets for public use, the distribution of mobile phone data applications, machine learning methods, and managerial implications.

### 6.1. Public Datasets in Mobile Phone Data

As Table 6 states, in the literature, many studies have made their datasets privately available, especially mobile phone data at the individual level (CDRs), due to privacy concerns, while open-source public datasets are limited and mostly suitable for specific applications related to human mobility patterns due to the absence of communication information.

### 6.2. Distribution of Mobile Phone Data Applications

Research on mobile phone data has been widely used in various applications in different domains because even the smallest bit of information is enough to trigger many new applications [1]. Although some studies on mobile phone data are used to develop their unique applications [99], most studies share a relatively broad classification. Figure 2 illustrates different applications of mobile phone data.

The data used in this study were collected from the Scopus database, and VOSviewer software was used to obtain a visual representation of the data. In the network presented in Figure 2, there are three clusters. The first cluster, the red cluster, contains nodes that represent applications derived from human activities and mobility in urban sensing, such as urban transportation, travel behaviors, urban dynamics, estimating population densities, and estimating origin–destination matrices. The nodes in the second cluster, the blue cluster, represent human communication behaviors, and social activities, such as social networking, mobile networking, 5G mobile communication, crime networks, churn prediction, and anomaly detection. The third cluster, the green cluster, contains nodes related to epidemiology and public health, such as health risk assessment, estimating physical distancing during pandemics, COVID-19, disease control and prevention of disease transmission, estimating air pollution, and estimating individual exposure to air pollution. We noticed some new and emerging applications in which mobile phone data have been used to monitor physical distancing and model the spread of COVID-19 for disease control and prevention.

### 6.3. Methods

Table 7 shows that various machine learning models and algorithms have employed mobile phone data to classify, cluster, and predict human communication behaviors and mobility patterns in various contexts. Most studies have used machine learning approaches to solve classification and clustering problems. Machine learning (ML) is a subset of artificial intelligence that can learn from data and make decisions with little or minimal human intervention [100]. On the other hand, deep learning (DL) is a subset of machine learning that can automatically learn useful features and representations from raw data and output results without human intervention [101]. However, DL algorithms require significantly more data than ML algorithms to function properly. Due to its complex multi-layer structure, a DL system requires a large dataset to eliminate fluctuations and produce high-quality interpretations.

As the table illustrates, most problems in mobile data are regarded as classification and clustering problems; for example, the churn prediction problem has been solved with supervised classification algorithms such as SVM and RF, while the problem of detecting anomalies in mobile phone data has been mostly solved with unsupervised clustering approaches such as k-means clustering.

In addition, supervised algorithms such as RF and SVM are the most widely used in the literature, and at 28 times, this is the highest share among all supervised algorithms. Meanwhile, k-means was the most frequently used unsupervised algorithm, as shown in Figure 3a. On the other hand, deep learning algorithms such as CNN and MLP are the most frequently used for classification problems, including the prediction of customer churn [131].

However, because mobile phone data are not labeled, the results above do not reflect that the supervised methods are suitable for other problems, such as land use classification and transport mode estimation, where semi-supervised learning techniques are used to tackle the lack of ground-truth data by relying on small subsets of labeled data, such as in [8,53,94,102].

Finally, some studies combine two or more algorithms to improve the classifier; in this approach, one algorithm works as a preprocessing mechanism to create a user profile, and the second algorithm is used to take these generated user profile clusters as input to the SVM classifier, as in [128,130]. Another example is studies [122,124], which use multiple algorithms, where unsupervised algorithms such as k-means and PCA are used for feature dimensionality reduction and supervised algorithms such as SVM and RF are selected according to which will achieve the best classification results.

### 6.4. Managerial Implications

This section presents some of the significant managerial implications of using mobile phone data.

Legal and ethical implications: Mobile phone data have benefited our daily lives in many ways, including preventing and controlling infectious diseases and assisting in the fight against criminal activities. However, mobile phone data are subject to privacy breaches because it contains sensitive information about individuals, such as the location of their homes, their most commonly contacted numbers, and their most frequently visited locations. This raises privacy concerns and ethical questions about using such data. Taylor [133] discusses some theoretical and practical implications of using mobile phone data, such as in situations involving a post-conflict state, where the ability to identify a group’s location, leaders, and communication networks may give hostile states a way to jeopardize the safety of the people whom they identify.

Future implications: The study’s implications revealed that spatiotemporal call volume features have been widely used to investigate the correlations between land use and human activities, to measure ambient population density, and to estimate population densities. They also found that there is not a lot of crime research literature about mobile phone data and that more crime applications, such as finding terrorist networks and credit card fraud, need to be found and studied.

## 7. Research Opportunities

The current state of mobile phone data in the context of detecting criminal activities and dynamics is still incomplete due to challenges with missing values and partial information (incomplete mobility and calling information). For example, [3,34,35,37] detected criminal behaviors by extracting criminals’ call features, such as outgoing and incoming calls, call frequency, maximum and minimum numbers of incoming or outgoing calls and messages, call timestamps, temporal changes in mobile phone call patterns, caller ID, and called ID, to detect criminal communication behaviors. On the other hand, [5,32,42] detected criminal mobility patterns by extracting spatiotemporal features such as variation over time, frequency, and distance. These features allow for the depiction of the mobile phone spatiotemporal patterns of criminals over a given period and allow for the measurement of the distance between the locations where the criminals visited and their homes. However, the results are still limited in providing a more comprehensive picture of all aspects of criminal behavior in terms of mobility patterns, communication behaviors, and social interactions. Consequently, there is a need to improve the detection of criminal behaviors by examining and considering the correlation of these analytical perspectives, which may ultimately provide different aspects of criminal activities.

Anomaly detection in mobile phone data is used to identify anomalies and outliers in the data by detecting anomalous activities (calling and mobility behaviors) or outliers’ calls in mobile cellular networks. The literature shows a number of anomaly detection applications that vary based on the characteristics extracted from mobile phone data, such as spatiotemporal features and call characteristics.

Phone fraud detection is one of the primary anomaly applications in mobile phone data. Fraud detection is used to detect fraudulent activities in telecommunications companies. For example, [128] and [130] detected fraudulent calls in mobile telecommunication networks based on extracting mobile subscribers’ calling behaviors. To depict subscribers’ calling behaviors, the authors extracted calling features from CDRs data, such as type of calls, call duration, frequency of a call, and call timestamp, which were later used to help classify mobile subscribers into three categories: genuine, fraudulent, and suspicious.

Another application of anomaly detection in mobile phone data is detecting emergency events or situations where the data are analyzed to detect population anomalies (abnormal population distribution), such as in [134,135], who estimated the crowd size (number of people in a given location) based on mobile phone usage volume. Here, detecting population anomalies is possible by detecting an abnormal increase in call volumes. Empirically, mobile phone usage or activity that is recorded at given locations based on a base station location where phone calls or SMS volume (high volume of incoming and outgoing phone calls and SMS messages) is much higher than in normal situations could be a sign of a disaster. Therefore, detecting population anomalies can help avoid situations, such as crowd disasters [135].

Another way to detect population anomalies or to detect anomalies in human behaviors was reported by [136], who extracted population trajectories represented by their spatiotemporal features (movement frequency) along with their call behaviors (total outgoing and incoming call volumes, call frequency). These features help depict locations or areas with a higher than normal movement frequency (abnormal movement frequency), which could be a sign of joyous events, such as Christmas Eve or New Year’s Eve gatherings. Unusually high call volume and movement frequency could also be a sign of a natural disaster, such as earthquakes or floods.

Other anomaly detection applications have been discussed in the literature, such as detecting anomalies in a cellular network by identifying sleeping cells and traffic overload [92,94,127]. Research on the current state of anomaly detection applications related to mobile phone data is still lacking. Thus, a systematic review is needed to fill in the gaps and investigate the current state of mobile phone data research on anomaly detection by discussing anomaly detection techniques and applications. Furthermore, many applications that are still lacking in the literature, such as malicious activity, credit card fraud detection, cyber intrusion, and terrorist activity, could be investigated.

Churn prediction is also an application that has been discussed widely in mobile phone data. For example, studies [137,138,139] have applied social network analytics methods (e.g., relational classifier (RC) and community detection algorithm) to predict customer churn, while machine learning techniques have been used in other studies [120,121,122]. In addition, multiple features have been extracted to depict customers’ calling behavior. For example, [120] extracted the number of outgoing calls and SMS, the number of incoming calls, the number of international calls and SMS, the total call duration, the number of incoming/outgoing MMS, and the data uploaded or downloaded volume per subscription. Similarly, [121] extracted the same features with additional features, including the use of the Internet, the average upload/download Internet access, and Internet usage for each customer per day. Therefore, it would also be beneficial to provide a survey that discusses the different features, different applications (such as the prediction for prepaid customers or postpaid subscribers in cellular telecommunication), and models used to predict customer churn.

Many studies have made their dataset privately available (especially mobile phone data at the individual level, known as CDRs data) due to privacy concerns, in which it contains sensitive details such as spatial–temporal trajectory and communication information about the receiving side of the communication as opposed to mobile phone data at the cell tower level, which does not reveal communication details, while publicly available datasets are mobile phone data aggregated at the cell tower level; thus, we encourage the need for a standard data collection framework in which it provides technical solutions as well as transforming mobile phone data remotely access to third parties by using the privacy-through-security approach, which will be available for interest researchers.

## 8. Privacy Concerns and Ethical Implications

This section discusses privacy concerns and ethical considerations about the fair use of mobile phone data. Due to the sensitive information that mobile phone data contains, it raises privacy and security concerns about its use for unethical purposes. The risk comes from the breaches of individual significant locations and the leak of their social network communications. Therefore, some technical, practical, and theoretical solutions have been provided to mitigate privacy concerns and ethical challenges.

For example, to mitigate ethical concerns, [140] proposed an ethical framework that defines ethical principles about using mobile phone data. These ethical principles are beneficence, respect for persons, justice, respect for the law, and the public interest. De Montjoye et al. [141] advised establishing an outside ethical committee to evaluate how mobile phone data are used.

However, most researchers have been exploring privacy-preserving techniques and tools that aim to mitigate privacy risks. For example, to maintain privacy and preserve individual and group privacy, [142] suggested applying differential privacy (DP) techniques to mobile phone data. DP protects privacy by injecting a desirable amount of noise into the sensitive data while maintaining a healthy trade-off between privacy and accuracy [143,144]. Pratesi et al. [145] proposed a framework called PRIMULE that can recognize risky user profiles in mobile phone data. The proposed framework aims firstly to mitigate the privacy risk of any set of profiles to which they apply the 𝑘-anonymity algorithm, a clustering algorithm that aims to ensure that each individual in a dataset cannot be distinguished from at least 𝑘 − 1 individuals whose information are also in the dataset. The second aim of PRIMULE is to avoid the misrepresentation of data (data distortion) during the analysis process in order to guarantee that the quality of the profiles is high in terms of similarity with respect to the original ones. Gramaglia et al. [146] proposed anonymization techniques to solve the privacy concerns regarding the disclosure of individual trajectories by applying generalization and suppression methods, one of the anonymity techniques that aim to achieve 𝑘-anonymity in the spatiotemporal trajectory of mobile phone data. The generalization of spatiotemporal information is carried out by minimizing the precision of trajectory samples in space and time, resulting in indistinguishability between the samples of two or more users. At the same time, suppression eliminates samples that are too difficult to anonymize from the trajectory information in mobile phone data.

Nonetheless, as those approaches are only providing anonymization solutions and are not providing a complete solution to all privacy and security concerns [140] and other issues related to ethics [141], there is still the possibility of re-identifying individuals [133]. Therefore, the privacy and security implications of mobile phone data still represent the most challenging barrier to the growing research effort [147]. Furthermore, without efficient techniques to resolve privacy issues and other ethical considerations, this is not going to change anytime soon [148].

## 9. Conclusions and Limitations

It is not surprising that mobile phone data have been used as a real-time sensor of human activities and dynamics. This is due to the characteristics of mobile phone data that record users’ interactions or activities on a mobile network, such as call and mobility activities, which can show how people move, socialize, and communicate. This in turn explains the explosion of mobile phone data in various disciplines and applications.

In this survey, we reviewed many applications driven by mobile phone data, such as urban sensing applications, crime applications, land use classification, anomaly detection, suspect classification, criminal networks, and social network applications. We also discussed several spatiotemporal and mobility features extracted from mobile phone data and differentiated them, explaining their functionality. We also shed light on open opportunities and challenges in mobile phone data. In other words, this study focused on three major subtopics: (1) the current state of mobile phone data applications; (2) techniques and methods used to predict and model human behaviors and mobility patterns; and (3) spatiotemporal and call features and characteristics extracted from mobile phone data to depict human activities and mobility patterns.

The present study aimed to investigate the current state of mobile phone data and its applications and to examine and explore human attributes and characteristics that can be derived from such data. In addition, this survey aimed to shed light on related studies or contributions made in mobile phone data research that the literature lacks, such as anomaly detection, churn prediction, and privacy concerns across various disciplines. Eight electronic databases are used for the retrieval of relevant papers. We filtered the results for articles published between 2013 and 2021 according to our inclusion and exclusion criteria. This allowed us to choose 148 articles for review before we synthesized them.

The results show that the spatiotemporal calling feature has been widely used to depict human behaviors. This feature allows for the estimation and calculation of the number of mobile phone devices or phone calls contacted or managed at a given cell tower at various temporal scales (e.g., hourly, daily, and weekly). This allows for the estimation of hourly dynamic population, the classification of land use, the estimation of ambient population, the investigation of the relationship between human dynamics and crime spatial–temporal patterns, defining the actual populations at risk, and others. The results also show that classification and clustering approaches have been widely used where algorithms such as SVM and k-means are used to classify or cluster human activities based on their calling or mobility features.

Although this study intends to survey mobile phone data methods and techniques in multiple domains, the results are not exhaustive, nor should they be considered a conclusive synthesis of all relevant studies. First, it only surveys contributions made between 2013 and 2021, with a primary focus on the areas of crime and urban research. Second, the study excludes academic content published in languages other than English.

To conclude, opportunities can also be found in mobile phone data, as newly developed methods may enable novel applications such as investigating relationships between human mobility patterns and infectious diseases such as COVID-19. Mobile phone data proved its effectiveness in estimating people’s travel and mobility patterns during the spread of infectious diseases and in measuring people’s daily mobility exposure to air pollution. Therefore, additional applications can be explored in the future.

## Figures and Tables

**Figure 1 sensors-23-00908-f001:**
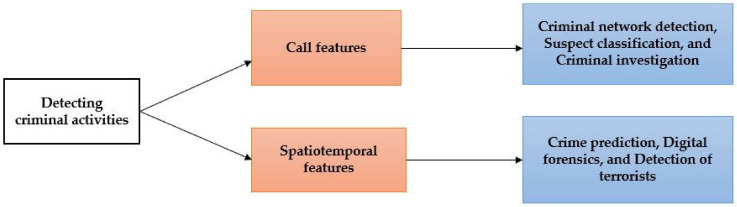
General concept of criminal activity detection.

**Figure 2 sensors-23-00908-f002:**
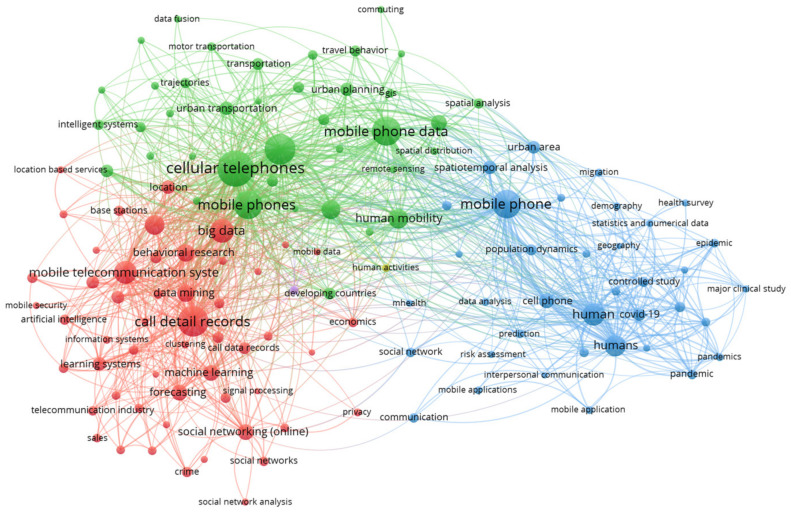
Network visualization of applications in mobile phone data.

**Figure 3 sensors-23-00908-f003:**
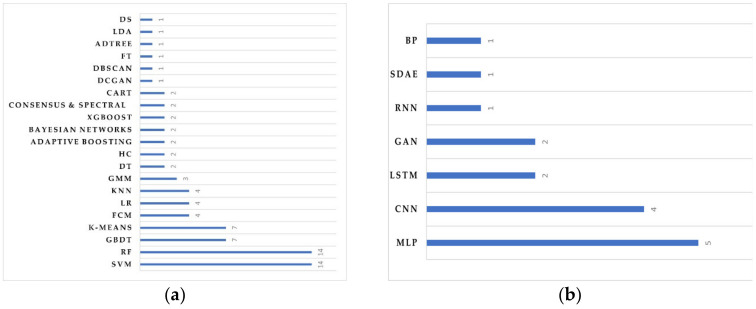
The distribution of machine learning algorithms (**a**) and deep learning algorithms (**b**).

**Table 1 sensors-23-00908-t001:** Prior research on human activities and land use based on mobile phone data.

Reference	Analytical Perspective	Feature/Characteristic	Application	Description	Algorithm/Technique
[8,9,11,12,13]	Human activities and mobility patterns	Spatiotemporal call volume: total and hourly call volume managed by each base transceiver station (BTS) (i.e., total number of calls or mobile phone devices managed by a given BTS over a given period)	Classification of urban land use types	These studies have depicted human activity patterns based on extracting spatiotemporal call volume features	FCM [8], NMF [9,13], SVM [11], and k-means [12]
[13,14,15]	Temporal changes in human activities	Temporal call patterns and volume: calculations or estimations of the number of calls or mobile phone devices managed by each BTS tower every hour in a seven-day week (i.e., weekdays and weekends)	Land use detection	These studies have detected land use patterns based on temporal changes in human activities to capture human behaviors’ variation over time (e.g., human activity trough in the middle of the day on weekends)	NMF [13], community detection algorithms [14], and latent Dirichlet allocation [15]
[26]	Human dynamics	Spatiotemporal features: cell tower identification that shows BTSs’ exact location and aggregated mobile network traffic activity for each BTS at 10-min time intervals	Investigation of relationship between human dynamics and land use	This study investigated the correlations between land use and human dynamics, depicting human dynamics as a graph in which nodes are BTS towers and edges represent communication traffic between two nodes	Community detection algorithms
[25]	Human commuting patterns	Users’ daily trajectories based on spatiotemporal features: users’ location represented by cell tower location (e.g., a residence location can be identified based on the most frequently used cell tower locations during the night hours)	Clarification of relationship between commuting flows and variables such as industrial, commercial, residential, and educational land use	This study’s main goal was to gain a fuller understanding of the relationship between land use variables and commuting flows, so a gravity model was used (i.e., a widely used technique for assessing commuting flow patterns), which shows that commuting between two locations i and j, with origin population mi and destination population mj, is proportional to the product of these populations and inversely proportional to a power law of the distance between them [25]	Gravity and regression models
[23]	Human daily and weekly activity patterns	Spatiotemporal call features: spatiotemporal call volume (i.e., total call volume and compared call patterns), communication habits, weekly patterns, and contact features	Land use classification	This study focused on various features to capture many aspects of human activity patterns and depict variation in human activities on weekdays and weekends	RF

**Table 2 sensors-23-00908-t002:** List of applications, methods, and features used in criminology based on mobile phone data.

Reference	Analysis Perspective	Feature/Characteristic	Application	Description	Algorithm/Technique	Geographical Unit/Spatial Unit
[29,30]	Human mobility patterns	Spatiotemporal features: cell tower IDs and timestamps to calculate the total number of mobile phone devices in each cell tower every hour	Crime prediction	Human mobility patterns extracted from mobile phone data can be used to predict crime hotspots	RF	Cellular network cells: 124,119 cells
[41]	Human mobility patterns	Spatiotemporal features such as cell tower IDs and timestamps to estimate footfall count entries in each cell per hour	Crime prediction	The results show that the relationship between crime activities and the diversity of the ages and ratios of visitors negatively correlated	Correlational analysis: Tjostheim’s coefficient	Grid cells: the geographic area is divided into 23,164 grid cells.
[42]	Human daily mobility patterns or daily population mobility patterns	Extracted spatiotemporal features: cell tower IDs and timestamps	Crime prediction	The daily mobility flows of the general population have been captured to provide a template of the daily mobility of criminals	Regression analysis: conditional logit discrete choice models	Census units: 1616 census units
[33]	Human mobility patterns and social activities	Spatiotemporal features and call logs: cell tower IDs, timestamps, and the number of phone calls or short message service (SMS) made and received	Crime prediction	Mobile phone data have been used to measure the ambient population at risk, and results showed a strong correlation between ambient population and criminal activities	Correlation analysis: Moran’s I statistic, and regression: negative binomial regression analysis	Grid cells: the study region is partitioned into equally sized grid cells of (306 × 306 m).
[28]	Mobile phone activity	Spatiotemporal features: timestamps and cell tower IDs to estimate or count the number of times a mobile phone device communicates with the cell tower, which this parameter has later used to measure the size of the ambient population	Crime prediction	The results showed strong correlations between the ambient population measures (workday population, mobile phone data, and population 24/7 daytime estimates) and crime patterns (the crime of theft from person)	Correlation analysis: Spearman’s rank correlation coefficient [ρ] statistics	Lower super output areas (LSOAs): cellular network grid cells converted to LSOA geographical units
[31,43]	Mobile phone activity	Spatiotemporal features: timestamps and cell tower IDs to calculate the total number of mobile phone devices in each cell tower every hour over a 3-month period	Crime prediction	A stronger correlation was found between ambient population and crime rates	Correlation analysis: Pearson correlation coefficient and point-biserial correlation coefficient	Grid cells of 200 × 200 m
[27]	Human mobility patterns	Spatiotemporal features: timestamps and cell tower IDs	Crime prediction	The results demonstrate a negative relationship between ambient population and street robbers’ criminal activities, in which ambient population has a significant effect by reducing opportunities to commit crimes	Correlation and regression analysis: discrete choice models and negative binomial regression	The geographical areas were created using Thiessen polygons, where 52,026 cell towers were mapped onto polygons
[44,45]	Intra-daily mobility patterns of the population	Spatiotemporal features: timestamps and cell tower IDs to identify the origin and destination of each user	Crime prediction	These studies proposed a new measure in calculating crime rates and exploring crime patterning, which is the exposed population at risk, which includes a mixed population of, for example, criminals, victims, and guardians. The results showed that the exposed population is more significant than the ambient population in exploring violent crimes in public spaces	Correlation analysis: Spearman’s rank correlation coefficient (ρ) statistics [44]. Regression analysis: negative binomial regression model (NBM) [45]	Lower super output areas: 1673 LSOAs
[32]	Daily movement patterns of migrant and native offenders	Spatiotemporal features extracted: timestamp and cell tower ID to count the number of mobile phone devices connected to a given cell tower on a per-hour basis. This feature helps to estimate ambient population and criminal movements when a crime takes place	Detecting criminal mobility patterns	The results show that the ambient population has a positive relationship with dynamic patterns of violent crimes committed by migrant offenders	Descriptive statistics and negative binomial regression models	The geographical areas were shaped using the Thiessen polygon technique, where 52,026 cell towers were represented as Voronoi cells
[5]	Spatiotemporal mobility patterns of terrorists	Spatiotemporal features of terrorists: (1) variation over time: this feature depicts the mobile phone spatiotemporal patterns of criminals over a given period and then allows to distinguish criminal mobility patterns and activities that contain varied locations over time (2) frequency: number of times a given cell tower has visited or contacted by criminals (3) distance: the distance between the locations where the criminals visited and their homes	Detecting mobility patterns of terrorists	This study identified the meaningful places for criminals based on the digital traces they left at home and other visited locations. The traces were then analyzed to determine the changes in the terrorist’s spatial behaviors	Correlation analysis: Spearman’s rank coefficient (ρ) statistics, Pearson’s correlations, and statistical analysis: the cumulative distribution function	Cellular network cells: cell tower locations were spatially approximated to the postcode area, which in the United Kingdom covers a small area of approximately 0.14 km^2^.
[3,34,35,36,37,46,47]	Criminal communication behaviors	Call features: outcoming/incoming calls, call frequency, maximum and minimum numbers of incoming or outgoing calls and messages, call timestamps, temporal changes in mobile phone call patterns, caller ID, called ID, type of communication (phone call, SMS, MMS, or voice), and call duration	Detecting criminal networks	These studies built multiple forensic systems to detect criminal networks based on their calling characteristics. Here, a criminal network is represented by a set of nodes (criminals) and the edges or links between them represent a communication (i.e., a phone call or SMS)	Social network analysis tools and graph algorithms such as Prim’s minimum spanning tree algorithm [35], the Girvan-Newman algorithm [34], Space algorithm [3], Blondel’s community detection algorithm [4], and Fruchterman–Reingold algorithm [47]	N/A Missing location data (i.e., the geographical position of nodes is unknown)
[48,49,50,51,52]	Communication and mobility patterns of suspects	Spatiotemporal and calling features: the SIM numbers and location ID of the suspects, calls made between the suspects, maximum call duration, call frequency, phone calls made at the crime location, the most frequent caller, the number of times the suspect called other suspects, suspect trajectories, and others	Identifying suspects and their associates	These studies built a call detail record query system to detect suspects and suspicious groups.	Big data technologies and analytics such as Hive, Hadoop MapReduce, and the Hadoop Distributed File System	The coverage areas of cell towers have not been intersected with any geographical units.
[4,39]	Suspects’ communication behaviors	Call features: call duration between suspects, maximum and average call duration, maximum duration of outgoing and incoming calls, standard deviation duration of incoming calls, phone calls made at the crime location, and others	Suspect classification	These studies built suspect classification models based on machine learning approaches that can classify suspects from non-suspects	Bayesian network [39] and graph convolutional networks [4]	N/A

**Table 3 sensors-23-00908-t003:** List of applications used in public health.

Reference	Application	Feature	Study Finding	Analysis Perspective
[62]	Dynamic estimation of individual exposure to air pollution.	Spatiotemporal	The results show that exposure to nitrogen dioxide (NO_2_) goes up by 4.3% during the week and by 0.4% on the weekends. Due to the fact that, during the week, people who live in small towns near big cities are exposed to more NO_2_ because they work in these cities	People’s travel patterns
[63]	Estimation of human exposure to air pollution	Spatiotemporal	The results indicate that the home-based method (HBM), which assumes that all individuals spend the entire day at their homes (individuals who are not highly mobile), is still a useful measure for estimating their exposure to air pollution	Daily mobility patterns
[64]	Controlling the spread of dengue fever	Spatiotemporal, SMS	The results show that the spatially targeting SMS policy that encourages people to avoid and cancel trips to high importation risk areas can help to reduce the risk of disease spread	Spatiotemporal travel patterns
[65,66,67]	Controlling and measuring the spatial spread of malaria	Spatiotemporal	These studies prove that mobile phone data are effective in controlling and estimating the spread of malaria parasites by analyzing the mobility patterns of individuals in countries such as Bangladesh, South Africa, and Madagascar	Human travel patterns
[60]	Investigation of the correlation between mobility patterns and COVID-19 cases	Spatiotemporal	The results indicate that a decrease in human movement (a reduction in the number of individual trips) is associated with a decrease in the growth rate of COVID-19 cases	Human daily mobility patterns
[68]	Real-time predictions of human movement during the Tokyo earthquake	Spatiotemporal	The proposed assimilation method yields encouraging results for estimating the real-time movement of people during earthquakes	Real-time human movement

**Table 4 sensors-23-00908-t004:** List of transportation applications in mobile phone data.

Reference	Application	Description
[75]	Traffic congestion detection	The study aims to classify traffic conditions into high, medium, and low traffic levels based on handover records that show the number of handover events in a cellular tower
[76]	Origin–destination flow estimation	The study aims to study traffic flow in Dhaka city by constructing origin–destination (OD) matrices based on phone users’ trajectories
[77,78,79]	Urban planning and management	These studies aim to investigate and understand the dynamics of human mobility and human travel patterns in urban areas, which paves the way for improving traffic planning, public transport design, and transportation infrastructure design
[19,80,81]	Transport mode detection	These studies analyze the travel behaviors of passengers in order to identify the modes of transportation that passengers take, such as metro, train, car, and bus

**Table 5 sensors-23-00908-t005:** Summarizes social network applications, algorithms, and network metrics.

Reference	Application	Feature	Algorithm	Network Measure
[34,47]	To detect criminal networks	Call features	Girvan–Newman and Fruchterman–Reingold	Degree centrality, eigenvector centrality, closeness centrality, transitivity, betweenness centrality, and transitivity
[3,35]	To detect criminal networks	Call features	Concept space approach and Prim’s algorithm	Vertex-centric, edge-centric
[84]	To detect customers who are likely to fail to pay their mobile bill	Call feature and spatiotemporal features	SLPA	Closeness centrality and reciprocity measures
[85]	To detect users’ social interactions	Call feature	Bron and Kerbosch’s	Persistence, disparity, and reciprocity measures
[86]	To detect ethnic communities in Ivory Coast	Call and spatiotemporal features	Louvain	Asymmetries and assortativity coefficient
[87]	To detect human spatial interactions in China	Spatiotemporal features	Infomap, Louvain, and REDCAP	Degree, strength, rich-club coefficient, and assortativity coefficient
[88]	To detect socio-economic groups in Ivory Coast	Call and spatiotemporal features	Louvain	Rich-club coefficient and PageRank
[26]	To detect he spatial interactions of communities in Milan, Italy	Spatiotemporal features	Louvain	Betweenness centrality, degree centrality, and PageRank
[89]	To detect individual ‘s spending behaviors	Spatiotemporal features	Louvain	Diversity, radius of gyration, and homophily
[90]	To detect socio-economic communities in Santiago, Chile	Spatiotemporal features	Louvain	Segregation measures (i.e., isolation metric)
[91]	To detect urban communities in Dublin, Ireland	Spatiotemporal features	Louvain	Newman’s modularity metric

**Table 6 sensors-23-00908-t006:** List of publicly available datasets used in the literature.

Dataset	Description	Application	Limitation/Link
**Nodobo mobile phone records dataset**	Nodobo contains mobile phone records of 27 graduating high school students from September 2010 until February 2011. These data were collected by a group of researchers at the University of Strathclyde, United Kingdom	Used in applications regarding detecting criminal networks [3,48] and anomaly detection [92]	It contains only communication data; hence, it is suited for applications related to human communication behaviors (https://pureportal.strath.ac.uk/en/datasets/nodobo-mobile-phone-usage-data (accessed on 20 September 2022))
**Smartsteps dataset**	Anonymized and aggregated human behavioral data derived from Telefonica Digital Company, Portugal.	Used in applications regarding crime prediction in urban areas [29,30,41]	It contains only spatiotemporal data (https://tis.smartsteps.telefonica.com (accessed on 10 August 2022))
**Telecom Italia dataset**	This mobile phone data published by Telecom Italia in 2014	Used in applications regarding land use detection [22], urban hotspot detection [93], anomaly detection [94], and mapping population density [95]	It Is only suitable for limited applications (http://www.telecomitalia.com/tit/en/bigdatachallenge/contest.html (accessed on 12 November 2022))
**Orange’s ‘Data for Development’ (D4D)**	This dataset contains two types of mobile data, mobile phone data at the aggregated level (aggregated CDRs) and the cell tower level. These data were generated from Orange mobile phone operator in Ivory Coast	Used in applications regarding home/work detection [96], land use detection [9], identifying user habits [97], and unusual event detection [21]	Only suitable for limited applications (http://www.d4d.orange.com (accessed on 21 October 2022))
**OpencellId database**	This is the largest open database of cell towers in the world	Estimating ambient population [98]	It contains only location data (https://opencellid.org/ (accessed on 14 November 2022))

**Table 7 sensors-23-00908-t007:** Machine learning (ML) and deep learning (DL) models and algorithms used by previous researchers in mobile phone data studies. The abbreviation of algorithms is shown in the back matter.

References	Algorithm/Model	Objective
[102]	SVM, NB	To classify user relationships
[8]	FCM	To classify urban land use in Singapore
[4]	GCN	To classify criminals from non-criminals
[39]	BN	To classify suspect users from non-suspect users
[103]	GBDT	To detect significant locations in users’ visiting patterns
[29,30]	RF	To classify geographical areas into two classes, high or low crime levels
[7]	RF	To predict population density in Portugal and France
[23]	RF	To classify urban areas in Tel Aviv
[104]	DBSCAN, GMM	The DBSCAN algorithm is used to cluster users’ trajectories into meaningful places, while GMM is used to identify users’ habits
[11]	SVM	To classify urban land use in Beijing into six classes, (a) residential, (b) business, (c) scenic, (d) open, (e) other, and (f) entertainment
[12]	K-means	To identify urban functional areas (UFAs) in Beijing
[105]	GAN	To create artificial maps of population density distributions
[106,107]	K-means	To classify city users based on their calling behaviors into different types of city geographic areas, including residents, visitors, and commuters
[108]	MLP	To predict the real estate price in Budapest, Hungary
[109]	RF, GBDT, SVM, Adaptive boosting	To reconstruct individual trajectories
[110]	MLP, CNN, LSTM	To predict crowd distributions of people in urban areas
[111]	NB, LR, RF, DT, KNN	To prompt or recommend the best mobile phone contract services based on customer communication behaviors
[112]	BP	To estimate individual exposure to particulate matter (PM2.5) air pollution
[113]	ADTree, FT, RF	To detect subscriber identity module box (SIMbox) fraud
[114]	LR, SVM-Linear, SVM-RBF, KNN, RF	To predict demographic features such as age and gender
[115]	SVM-Linear, Logistic regression	To predict demographic features such as age and gender
[116]	NB, SVM, DS, RF, RNN	To predict the next location of tourists
[117]	HC	To cluster human mobility patterns based on similar individual trajectories
[118]	GAN	To generate synthetic data of mobile phone data
[119]	KNN, RF, SVM-Linear, SVM-RBF+CNN, LSTM, SDAE	To construct a classifier that enables the recognition of fraudulent phone calls
[120]	RF, GBDT, SVM +CNN	To classify churner customers from non-churner customers
[121]	DT, RF, GBDT, XGBoost	To predict customer churn in Syriatel telecom company
[122]	MLP, SVM, Bayesian networks	To detect prepaid customer churn in mobile telecommunications companies
[123]	RF, DT, MLP, GBDT	To build predictive models that can classify customers into different categories of loyalty, such as very high value customers (greater loyalty), medium value customers (average loyalty), and others
[124]	LDA, SVM-RBF, XGBoost), RF, LR, NB, KNN, Bagged CART, CART, GBDT, C5.0	To predict customer demographic variables such as age and gender in Syriatel Telecom Company
[125]	K-means, DBSCAN	To detect fraudulent calls in telecommunications companies such as
[126]	GMM, ANN	To build a clustering-based classification model to classify cellular network traffic patterns into high-activity area, medium-activity area, low-activity area, etc.
[92,94,127]	K-means, GMM+CNN	To detect anomalous behavior through the identification of anomalous activities of mobile phone subscribers [92], to detect anomalies in a cellular network such as sleeping cells or unusual high call volume in a given region (traffic activity) [94]
[128]	FCM	To classify mobile subscribers based on extracting their calling features into three classes genuine, fraudulent, and suspicious
[129,130]	HC, k-means, FCM, SVM	To detect fraudulent behaviors in telecom companies such as detecting fraudulent calls
[10]	K-means, FCM, spectral clustering, consensus clustering	To cluster land use in Madrid
[131]	FKNN, MLP, C4.5, SVM GBDT, LR, RF, Adaptive boosting	To classify mobile customers into two classes, churners or non-churners
[132]	K-means	To cluster users according to their weekly mobility patterns into six different profiles

## Data Availability

Not applicable.

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
