# Peer review of "Mobile Phone Data: A Survey of Techniques, Features, and Applications"

_sensors, 2023, doi:10.3390/s23020908_

Round 1

Reviewer 1 Report

Thank you for the opportunity to read this paper. The paper reviewed methods and analytic approaches applied for assessing and predicting human mobility and activity patterns from mobile phone data.  The review was clearly written and English was of a high standard. Section 2, was particularly strong providing a great description of the two broad types of mobile phone data commonly used in research. However, while the review set out to establish the use of mobile phone data across various disciplines and domains the researchers only focus on landuse classification and crime. This presents a very incomplete picture of the current state of the literature.  Mobile phone data has been used extensively in transport and health research yet this is not featured in the review. In fact transport studies utilise mobile phone data more extensively then other fields and this should be depicted in the review.

This review fits within the scope of the journal and the special issue, however, there are many reviews of the use of mobile phone data for research. Most of the existing reviews are specified to a field or discipline eg transport, health. However, there was an excellent review of the use of mobile phone datasets for research conducted in 2015 please see: Blondel, V.D., Decuyper, A. & Krings, G. A survey of results on mobile phone datasets analysis. EPJ Data Sci. 4, 10 (2015). https://doi.org/10.1140/epjds/s13688-015-0046-0

If this paper was a broad overview of ALL domains and disciplines and it was an update on the 2015 review, it would be a great addition to the literature. Currently, this paper brings to the literature very detailed information about the application of mobile phone data in the criminological setting. This is still an addition to knowledge as such a review has not been done before. However, the manuscript would need to be re-titled and re-framed if the aim was to review the application of mobile phone data in criminology.  

The review is well researched and the literature that the authors cite is extensive. The tables present an excellent summary of the methodological approaches taken to detect landuse classification and crime but the full range of mobile phone data possibilities is not captured.

The areas noted for future research in section 6 are interesting but do not appear to closely align with the review. I would suggest additional thinking around identification of current gaps in the research on mobile phone data use and its application. In particular a section should discuss concerns about balancing the ethical requirements to maintain privacy alongside the needs to progress research.  

Reviewer 2 Report

The authors aim to provide a survey of different methods and approaches for assessing and predicting human communication behaviours and mobility patterns from mobile phone data and differentiate. them in terms of their strengths and weaknesses. Overall, the paper is well-written, and only minor changes are needed. 

1. In the introduction, please, elaborate on the similar literature reviews, and discuss how your work is different. 

2. In the abstract, clearly state your work's methodology.

3. There should be a chapter on Methodology in which the author should elaborate on the methodology used. As it seems they are using a non-systematic literature review. See more information about it at (2019). Planning, Conducting and Communicating Systematic Literature Reviews. Journal of theoretical and applied electronic commerce research14(3), 1-4.

4. Please ensure that the abbreviations are used consistently in the paper. Please, show the abbreviation when you mention the term for the first time in brackets, and then consistently use it in the paper.

5. Many abbreviations of the algorithm are presented; it would be a good idea to present them in the paper's appendix. 

6. Better elaborate on the difference between machine learning and deep learning.

7. Remove the bullets from the paper. Chapter 6 cannot consist only of the bullets.

In the last section, please focus on “Discussion, Implication, and Conclusion” to include (1).     Summary of the research - what was the goal, and how was it attained (2).     Discussion of why the authors found these results and how they comply (or not) with the Literature Review. (3).     Managerial Implications (4).     Limitations of the paper (5).     Future Studies and Recommendations  

Round 2

Reviewer 1 Report

The authors have incorporated all suggested revisions. I have no further feedback. Well done on an informative and interesting piece of work.